# Canine Electroencephalography Electrode Positioning Using a Neuronavigation System

**DOI:** 10.3390/ani14111539

**Published:** 2024-05-23

**Authors:** Casey Beatrice Rogers, Sebastian Meller, Nina Meyerhoff, Holger Andreas Volk

**Affiliations:** 1Department of Small Animal Medicine and Surgery, University of Veterinary Medicine Hannover, 30559 Hannover, Germany; casey.beatrice.rogers@tiho-hannover.de (C.B.R.); sebastian.meller@tiho-hannover.de (S.M.); nina.meyerhoff@tiho-hannover.de (N.M.); 2Center for Systems Neuroscience Hannover, 30559 Hannover, Germany

**Keywords:** EEG, neuronavigation system, electrode position, cortical regions

## Abstract

**Simple Summary:**

Simple Summary: This study investigated the current methods of placing electrodes used to measure the electrical activity of the brain in dogs. Recording the brain’s electrical activity can be of great importance, for example, in dogs affected by epilepsy. Electrodes are placed manually, by orienting oneself on landmarks of the skull. This study investigated this positioning using a 3D cranial navigation system and found that manual electrode placement leads to a high variance in electrode positions. Unprecise electrode positions could influence the recording quality of the brain’s electrical activity, as different electrode positions also record different areas of the brain. A cranial navigation system can be used to place electrodes precisely. This can be of importance when using recordings in research settings, when recordings of particular areas of the brain are required, or when the electrical activity of the same individual is recorded in repeated sessions and these recordings should be comparable.

**Abstract:**

Background: Studies in people suggest that surface electroencephalography (EEG) electrode positions vary across participants and that the consistency of these positions is electrode-, region-, and examiner-dependent. The aim was to investigate the variability in EEG electrode positions to their underlying cortical regions (CRs) in dogs using a neuronavigation system and evaluate the use of said system in electrode positioning, via a cadaver study with 22 dogs. CT scans and MRI were performed for each dog. These were uploaded onto a neuronavigation system where the desired CRs were annotated. The electrode positions were marked on the heads, which were positioned using only a previously established guide and anatomical landmarks. Using the neuronavigation system, alignment or deviations from the desired CRs were noted. Fifty-three percent of all the marked electrode positions showed an alignment with the desired CRs. Thirty-three percent showed no alignment, and fourteen percent showed partial alignment. Three percent deviated to different cortical lobes. Placement via the neuronavigation system enabled reliable and replicable electrode positioning and CR alignment. The standard for EEG electrode placement in dogs is subjected to a high variance. A neuronavigation system can aid in more precise electrode placements. Specific gyri cannot accurately be evaluated on EEG without imaging-controlled electrode placement.

## 1. Introduction

Electroencephalography (EEG) has become a useful tool in the diagnostic approach for canine idiopathic epilepsy (IE) and is required for a Tier 3 diagnosis [1]. It aids in differentiating IE from other diseases such as canine paroxysmal dyskinesia (cPD) and structural epilepsy and diagnoses non-convulsive status epilepticus [1,2,3,4].

In human medicine, the standard EEG for clinical use consists of 16 to 20 channels using a longitudinal bipolar, transverse bipolar, or referential montage with a 10-20 system [5]. Additionally, the 10-10 system, with more than 70 electrode positions and the ability to display up to 256 channels, is used in clinical EEG [5]. Besides diagnosing epilepsy in people, scalp electrodes can be used for brain mapping, which can identify and differentiate normal brain function and abnormal activity [6]. Scalp or invasive cortical surface electrode EEGs are also used to aid in identifying the area of the cortex from which clinical seizures are generated, i.e., the epileptogenic zone, the area of cortex which generates epileptic seizures (Table 1) [7]. Cortical and depth electrodes are used to identify potential surgical treatment options [8,9,10,11]. Additionally, EEGs are used for transcranial magnetic stimulation (TMS) in people and in experimental animal settings (rodents, cats, and dogs). EEG during TMS allows the control of the TMS parameters based on EEG activity [12,13]. TMS is a treatment option for epilepsy in people and dogs [14,15].

Various EEG montages have been described in the veterinary literature [4,18,19,20,21,22,23]. The montage used in this study is based on work from Pellegrino and Sica, in which a standardized EEG montage using specific anatomical landmarks was created to consider EEG needle placement in different canine skull types [22]. It is based on the 10-20 system used in people. Each electrode position was investigated by manual dissection. The brain-projection areas (BPAs) of the cortex and the specific gyri or fissures for each individual electrode were documented [22]. This canine montage has since been used for EEG studies with mild variations in the positioning of individual electrodes or additional electrodes [24]. Standard electrode positioning consists of 12 electrodes positioned as described and shown in Table 2 and Table 3 and Figure 1 and Figure 2.

Daniel et al. investigated the correlation of surface electrodes and the underlying cortical regions (CRs) in dogs comparatively using magnetic resonance imaging (MRI) and manual dissection in one case [25], showing that the location of the Fp, F, and P electrodes did not correlate with their underlying respective CR [25]. To the authors’ knowledge, no further studies have since been published that investigate the accuracy of using anatomical landmarks to position canine EEG electrodes. However, several studies in people suggest that electrode positions vary across participants and that the consistency of these positions is electrode-, region-, and examiner-dependent [26,27,28]. The studies have shown that the variability in the underlying CRs was smallest for frontal and temporal electrodes and greatest for central and parietal electrodes [12,26,29,30].

An unpublished study by Poma et al. aimed to investigate the accuracy of each electrode position with its underlying CR using a neuronavigation system and testing the application of the neuronavigation system for electrode placement [31]. In the current study, we further explored this observation by applying the EEG guidelines by Pellegrino and Sica (2004) [22] and used a novel neuronavigation system to compare the accuracy of the placement of the EEG electrodes to the targeted cortical area. We hypothesized that surface electrode placement using anatomical landmarks of the skull cannot accurately represent specific gyri, due to the high variability in canine head and brain shapes and potential structural abnormalities. Additionally, the limited spatial resolution of surface electrodes might complicate the exact representation of individual gyri.

The aim of the current study was to investigate the variability in EEG electrode positions to their underlying CR in individual canine cadavers using a neuronavigation system and evaluate if said system could be used for electrode positioning in dogs.

## 2. Materials and Methods

### 2.1. Study Design and Animals

A cadaver study was performed at the University of Veterinary Medicine Hannover. Twenty-two canine cadavers were used. Following their death or euthanasia, the dogs were donated to science by their owners, and written consent was obtained. Dolicho-, meso-, and brachycephalic dog breeds were used, including mixed-breed dogs (n = 4), French Bulldogs (n = 2), German Shepherds (n = 2), Labrador Retrievers (n = 2), a Labradoodle (n = 1), a Galgo Español (n = 1), a Flat-coated Retriever (n = 1), a Bearded Collie (n = 1), a Boxer (n = 1), an Elo (n = 1), a Shar Pei (n = 1), a Collie (n = 1), a Bernese Mountain Dog (n = 1), a Dogue de Bordeaux (n = 1), a Whippet (n = 1), and a Fox Terrier (n = 1).

The dogs were frozen at −18 °C prior to this study for storage.

### 2.2. Imaging and Neuronavigation Software

At the time of this study, the cadavers were defrosted for a time period of 27 to 38 h (median 30 h), depending on the size of the dog, for imaging. Computed tomography scans (Philips IQon Spectral CT (Philipps Medical Systems GmbH, Hamburg, Germany)) (CT) and 3-tesla MRI (3.0 Tesla MRI SmartPath to dStream for XR (Philipps Medical Systems GmbH, Hamburg, Germany)) of the heads were performed.

Both 3D T1-weighted images and 3D T2-weighted images were taken. The MRIs were assessed to rule out patients with severe structural changes in the brain. This was performed to avoid interference with the physiological brain anatomy, which the previous study for electrode placement is based on [22].

The images were uploaded onto a neuronavigation system (Neuronavigation CranialMap Stryker (Stryker GmbH & Co. KG, Duisburg, Deutschland)), and CT and MRI images were fused to create a 3D image of each dogs’ head. The system was used to annotate the BPA of each electrode, marking the gyri and fissures as described in Table 3. To aid the orientation, an atlas of the canine brain [32] and an MRI atlas [33] were used. These annotations were named using the correlating electrode terminology, as seen in Table 2. For instance, the left prorean gyrus was annotated as Fp1, the right prorean gyrus as Fp2, the left rostral part of the ectomarginal gyrus as P3, and the right rostral part of the ectomarginal gyrus as P4. This was carried out for all 12 electrodes.

### 2.3. Electrode Terminology and Placement

The canine head was placed and fixated on a table prepared for subsequent use with the neuronavigation system. Using the guide described in Table 3, the position of each electrode was marked on the shaved canine heads, using only the anatomical landmarks of the head (Table 3 and Figure 1 and Figure 2) and without the use of the neuronavigation system, CT, or MRI images. The used terminology can be found in Table 2.

The neuronavigation system was then set up, including a patient tracker fixated on the dog’s nose (dorsal part of the maxilla) via a bone screw and an instrument tracker to navigate the dog’s head and digitalize the electrode positions. The instrument tracker was used to digitalize the manually marked electrode positions. These were given double-digit numbers according to the respective electrode (e.g., Fp11 for electrode position Fp1, or P33 for electrode P3). To determine the optimal electrode position in correlation to the previously annotated CR, the instrument tracker was kept at a 90 °C angle to the brain surface and at the shortest distance to each CR and positioned until it correlated with the annotation for each electrode. These were named with triple-digit numbers, e.g., Fp111 for electrode Fp1 or P333 for electrode P3. This resulted in three digitized points on each head: the desired brain-projecting area (single-digit numbers), the location of the manually marked electrode position (MPE) (double-digit numbers), and the position of the neuronavigation system-guided marked electrode position (NSE) (triple-digit numbers).

### 2.4. Evaluation

The digitized points were then evaluated for alignment of the MPE with the desired BPA, the deviation to the BPA if no alignment was found, and what CR the MPE was overlying. Alignment was defined as the electrode’s position overlaying the desired BPA stated in Table 3. Partial alignment was defined as an electrode position aligning with the desired BPA in at least one of three MRI planes. No alignment was defined if the electrode position did not overlay the desired BPA or if no CR could be defined under that position. If a deviation was found, the direction (e.g., lateral, rostral, caudal) of the deviation compared to the ideal position for the electrode was additionally noted. In addition, the NSEs were compared to the MPE.

## 3. Results

### 3.1. Animal Data

The dogs’ weight ranged from 7 to 59 kg (mean 25.2 kg), with 11 (n = 11/22) male dogs and 11 (n = 11/22) female dogs. The age ranged from 1 year to 16 years and 3 months (mean 7 years and 8 months).

### 3.2. Alignment

Fifty-three percent (n = 139/264) of all MPEs showed an alignment with the intended BPA stated in Table 3 for each electrode position.

Fourteen percent (n = 38/264) of all MPEs showed a partial alignment with the BPA stated in Table 3 for each electrode position. 

Thirty-three percent (n = 87/264) of all MPEs showed no alignment with the BPA stated in Table 3 for each electrode position. This meant that the electrodes either did not overlay any CR or covered a different BPA than what is stated in Table 3. However, of the electrodes with no BPA alignment, an additional 3% (n = 8/264) showed no alignment with the associated cortical lobe mentioned in Table 3. The percentages of aligned, partially aligned, or deviated electrodes for the individual electrode positions can be found in Table 4. A good alignment of the NSE with the desired BPA was possible for all electrode positions. No deviation or partial alignment was seen in this approach.

### 3.3. Deviation

The deviated electrodes showed lateral, lateral and rostral, lateral and caudal, caudal, caudal and dorsal, and rostral and dorsal deviations. The individual deviations can be found in Figure 3 and Table 5.

### 3.4. Cortical Location

The detailed cortical structures that the manually placed deviated electrodes overlaid can be found in Table 6 or Appendix A for each individual electrode position and dog.

For 32% (n = 28/87) of the electrodes which did not align with the BPA, no underlying CR could be defined. Ninety-three percent (n = 26/28) of these electrodes were Fp electrodes. The nonaligned F electrodes diverted to the prorean gyrus and the postcruciate gyrus, or no underlying CR could be defined. The nonaligned P electrodes diverted to the medial part of the ectomarginal gyrus, the caudal part of the ectomarginal gyrus, and the precruciate gyrus of the frontal lobe and, therefore, additionally, to a different cortical lobe. The diversion of the O electrodes was to the medial part of the ectomarginal gyrus of the parietal lobe in all nonaligned electrodes. This caused a diversion to a different cortical lobe in all diverted O electrodes. The nonaligned T electrodes diverted to the ectosylvian gyrus, the ectomarginal gyrus, the sylvian gyrus, and the composite gyrus. Additionally, four of the T electrodes diverted to the occipital gyrus and, therefore, a different cortical lobe.

## 4. Discussion

Surface EEG can be used to localise epileptic activity to a certain brain area. However, anecdotal evidence suggests that surface electrodes are often not placed over the targeted cortical area. In the current study, the accuracy of EEG electrode positions to their underlying CR and BPA was investigated using a neuronavigation system. Additionally, this study examined whether positioning electrodes using the neuronavigation system improved accuracy. Overall, only half of the marked electrode positions were located above the targeted cortical area. The findings of this study show the highest variability in the FP, F, and T electrodes and the lowest to no variability in the P, O, and Cz and Oz electrodes. People studies have shown the opposite, with highest variability in electrodes in the central and parietal areas and the lowest in the frontal and temporal areas [26,29].

The placement of electrodes via the neuronavigation system enabled the reliable and replicable positioning of all the electrodes and good alignment with the required CR and BPA. The use of the neuronavigation system avoided the need for manually identifying landmarks, which could result in unreliable EEG assessments due to individual variations among patients and examiners [27]. Limiting factors for the use of the neuronavigation system were the size of the head and the length of the nose in the dogs used in this study. In this study, the patient tracker was fixated on the dorsal part of the maxilla via a screw, which was challenging in small-sized and brachycephalic dogs. However, an alternative dental fixation of the tracker was developed, using dental adhesives, which could be used in future patients who do not qualify for maxilla fixation.

The high prevalence of variance found in this study compared to previous findings [22,25] could be related to the difference in electrode position control. To the authors’ knowledge, this is the first study that investigates electrode positioning without manual dissection, using imaging technology in dogs. We suspect that manual dissection as a means of electrode position control does not account for changes in position caused by brain shift. Brain shift, a deformation of the brain, is a well-known phenomenon in neurosurgery. It can be caused by several factors directly or indirectly related to surgery, for example, including changes in gravity, head position, fluid drainage, or changes in intracranial pressure. Brain shift can range from a few millimetres to more than 25 mm [34]. This is a possible explanation for differences in our results compared to previous studies of electrode positions with variance up to a shift to a different cortical lobe. Additional factors not noted in previous studies for EEG positioning in dogs are age-related changes to the canine brain like cortical atrophy and ventricular enlargement [35], which potentially inflict on the relation to the surface position of an electrode and position of the underlying CR.

Especially in electrodes overlaying an array of brain structures, such as the P and T electrodes, the authors advise against specifying a definite BPA in terms of gyri und fissures for manually placed surface electrodes. Surface electrode recordings vary in accordance with their orientation and distance to the electrical activity that originates in neurons in the underlying brain tissue. The value recorded is distorted by intermediary tissues and bones. Hence, not all neurons contribute equally to an EEG signal, with an EEG predominately reflecting the activity of cortical neurons nearest to the electrodes on the scalp [36]. The ectomarginal gyrus (BPA for P electrodes) is in close proximity of the suprasylvian and marginal gyri, and the pseudosylvian fissure (BPA for T electrodes) is surrounded by the sylvian, ectosylvian, and ectomarginal gyri [32,33], making deviations of a few millimetres, susceptible to changes in the nearest cortical neurons. However, this does not impact the activity of the cortical lobe, as these summarize an array of specific structures in defined cortical areas [32], making diversions of a few millimetres within the area of the cortical lobe most likely irrelevant to the recording of activity generated by the entire lobe. Recording activity from a highly localized brain region requires the use of cortical surface or subdural electrodes. By eliminating distance and the distorting barriers, each electrode records the cortical area covered only by that electrode [7,37]. In people, the positioning of non-individualized surface electrodes has been discussed to be inappropriate when EEG or evoked potential techniques depend on accurate electrode placement [28], additionally limiting the comparability of longitudinal studies in the same patients, if the electrodes are not placed accurately [28]. This suggestion in people contradicts Pellegrino and Sicas’ statement that the manual placement used in this study “enables recordings to be made over the same brain areas at different times, even in animals with different skull types, so the electrical activity of the same anatomical area can be compared in successive recordings in the same animal or amongst different animals” [22]. However, due to the lack of recorded EEGs and the possibility of comparable follow-up EEGs, a correlation of this suggestion for EEGs in people to canine patients cannot be confirmed in this study.

### 4.1. FP Electrodes

The current literature states that “the employment of frontopolar electrodes (Fp1 and Fp2) allows the recording of electrical activity from the orbitofrontal area, even though the frontal sinus is in between the brain and the recording electrode” [22]. However, we could not establish an association to a CR in 59% of these electrodes, meaning that the electrode position was over the sinus in an area where no brain structure was underlying or in close scope to the respective electrode. Similar findings were seen in a previous study [25]. The dogs affected by this were all large-breed dogs or large dolichocephalic dogs. Since this was a postmortem study, no recordings from this area could be made to conclude whether the placement of these electrodes also interfered with the electrical evaluability. Compared to the dolichocephalic dogs or large-breed dogs, in small brachycephalic dogs we found, that the Fp electrodes overlayed the frontal lobe well; however, no distinguishment between the precruciate (BPA for F electrodes) and prorean gyri (BPA for Fp electrodes) could be made. In these dogs, good recordings of the frontal lobe should be possible; however, a distinction between the gyri might prove difficult.

### 4.2. F Electrodes

All frontal electrodes remained within the BPA of the frontal lobe, despite not all of them aligning with the precruciate gyrus. Although not being evaluated in this study, the frontal electrodes should allow reliable readings of the electrical activity of the frontal lobe. Nevertheless, an exact distinction between individual gyri might not be possible due to the individual variation and deviation shown for these electrodes.

### 4.3. P Electrodes

Due to the mid-brain position of the parietal electrodes, no partial deviations were found in these electrodes. Sixty-eight percent of P electrodes aligned with the rostral part of the ectomarginal gyrus. However, apart from the electrodes in one dog, all the other deviated P electrodes remained within the margin of the ectomarginal gyrus, just further caudally.

### 4.4. O Electrodes

The partially aligned occipital electrodes had a caudal or caudal and lateral deviation, placing them behind the caudal border of the brain (see example in Figure 4). Due to no other CR underlying these positions, this likely would not impact the recording of these electrodes, apart from potentially interfering with signal strength due to the higher distance to the BPA.

### 4.5. T Electrodes

As mentioned above, the T electrodes are subjected to a high variance in electrode position due to the close proximity of other brain structures and the narrow margin of the pseudosylvian fissure. Moreover, the results indicate that the placement of the electrodes is susceptible to a higher margin of error due to the limited orientation options available aside from the anatomical landmark. The T electrodes are not in affiliation with the electrodes placed dorsally on the head, and, therefore, these cannot function as additional orientation guides. This is supported by the asymmetrical distribution of unaligned electrodes, with three more deviated electrodes on the left side (T3) compared to the right side (T4), most likely caused by an error of the examiner which marked the electrode positions. However, similar to the P electrodes, the variance in the electrodes remains within the temporal lobe, with the exception of four electrodes, which each diverted to the occipital lobe. This diversion could be found in both of the French Bulldogs used in this study. However, due to the limited sample size of French Bulldogs in this study, it is unclear whether this observation is coincidental or potentially breed-related.

The number of cadavers available for this study provides a general limitation of this study, with 22 dogs of a high breed variety being unsuitable for advanced statistical tests and analyses. Additionally, due to the study design, no EEGs were performed to compare the changes in electrical activity in the diverted electrodes compared to their supposed position. In future studies with live animals, a comparative study should be obtained to evaluate whether a difference in electrical activity can be seen for diverted electrodes compared to the optimal electrode positions.

Nevertheless, the results showed that scalp electrode placement using anatomical landmarks of the skull cannot accurately represent specific gyri and fissures. It showed a high variability in EEG electrode positions to their underlying CR in individual canine patients and the unprecise placement especially of the temporal electrodes. The use of a neuronavigation system can eliminate factors such as variations in electrode position due to skull shape and brain shape (abnormalities) and faulty manual positioning due to a lack of orientational landmarks. Using neuronavigation system-guided EEG electrode positioning will most likely not be of daily clinical relevance due to the time it takes for the procedure and the need for advanced imaging; however, it may be a useful tool for specific purposes, such as the use with TMS, deep brain stimulation, epilepsy surgery, diagnosing the symptomatogenic zone, the irritative zone, or the seizure-onset zone, brain mapping, or in experimental settings, especially when using more than 12 electrodes. In human medicine, neuronavigation systems are used for guided electrode positioning and to implant cortical electrodes [37,38,39].

## 5. Conclusions

In summary, the current standard for electrode placement in dogs is subjected to a high variance in the exact location of each individual electrode pair, canine head shape, and each individual dog. A neuronavigation system can aid in a more precise electrode placement, compared to a manual electrode placement approach. Specific gyri and fissures cannot accurately be evaluated on EEG without imaging control, a multitude of electrodes, or subcortically placed electrodes. Nevertheless, the positions represented the desired cortical lobe in most cases.

## Figures and Tables

**Figure 1 animals-14-01539-f001:**
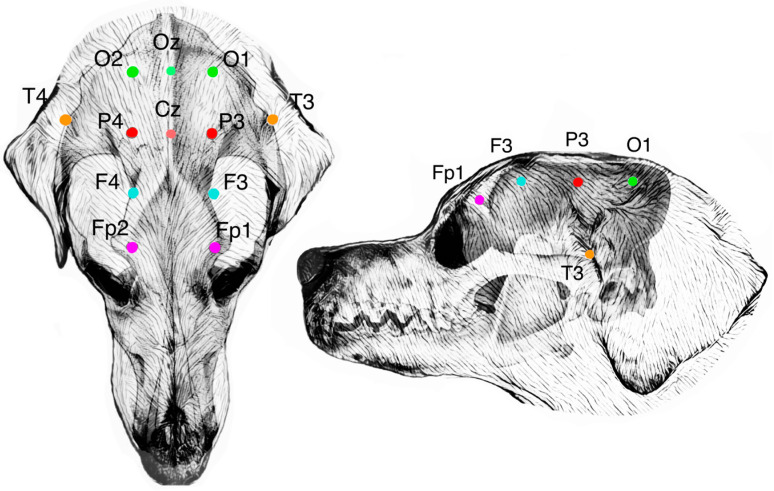
On the left: a dorsal view of a mesocephalic canine cranium showing the placement of the electroencephalography recording electrodes according to Pellegrino and Sica [22]. Fp, frontopolar electrode; F, frontal electrode; P, parietal electrode; O, occipital electrode; T, temporal electrode; Cz, central vertex electrode; and Oz, central occipital electrode. On the right: a lateral view of the left side of the mesocephalic canine cranium showing the placement of the left electrodes Fp1, F3, P3, O1, and T3.

**Figure 2 animals-14-01539-f002:**
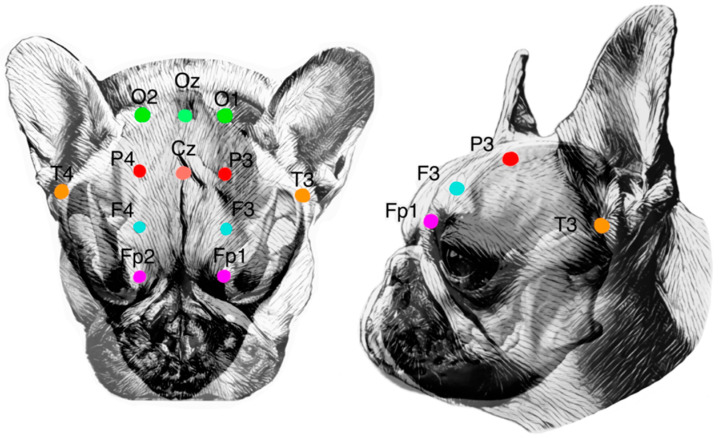
On the left: a dorsal view of a brachycephalic canine cranium showing the placement of the electroencephalography recording electrodes according to Pellegrino and Sica [22]. Fp, frontopolar electrode; F, frontal electrode; P, parietal electrode; O, occipital electrode; T, temporal electrode; Cz, central vertex electrode; and Oz, central occipital electrode. On the right: a lateral view of the left side of the brachycephalic canine cranium showing the placement of the left electrodes Fp1, F3, P3, and T3.

**Figure 3 animals-14-01539-f003:**
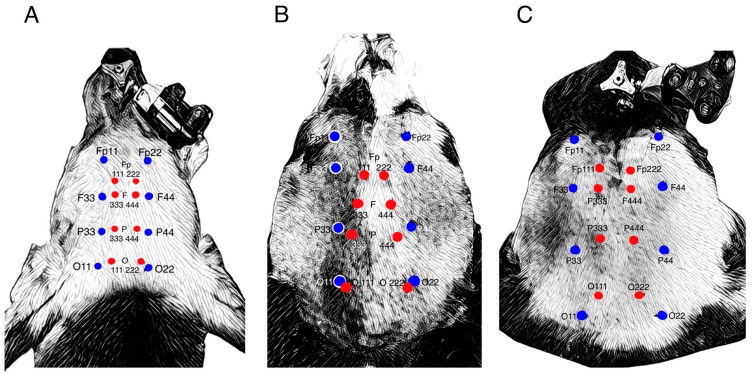
Showing the position of the manually placed electrodes (blue) and the electrodes placed using the neuronavigation system for accurate placement above the targeted brain-projection area (red). The used terminology is adapted from Table 2, with the double-digit numbers being the manually placed electrodes, and the triple-digit numbers’ electrodes being the ones placed via neuronavigation. (**A**) The surface position of the different placement approaches on a dolichocephalic dog (dog 5), (**B**) a mesocephalic dog (dog 15), and (**C**) a brachycephalic dog (dog 19).

**Figure 4 animals-14-01539-f004:**
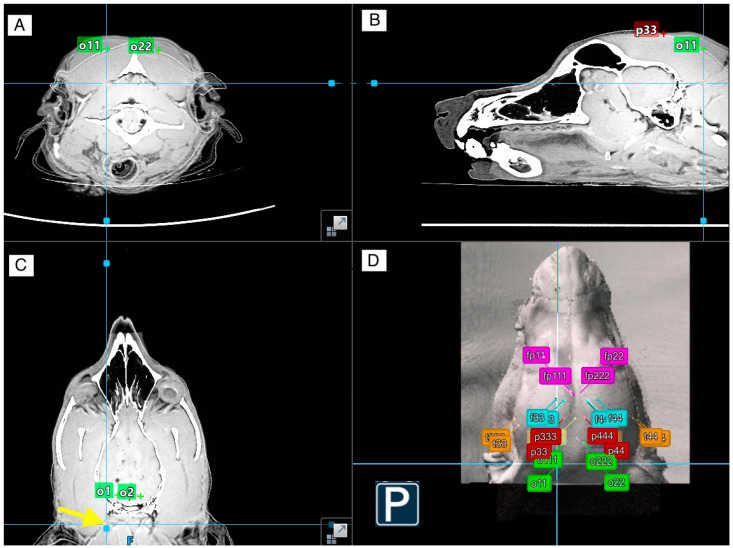
Showing the neuronavigation view of the manually marked occipital O1 (annotated as o11) electrode positions in (**A**) the transverse plane, (**B**) the sagittal plane, and (**D**) the 3D reconstruction of the head with all the annotated electrode positions in dog 6. (**C**) shows the dorsal plane as well as the annotated occipital gyrus (o1 and o2), and the yellow arrow pinpoints the position of the manually marked electrode position if followed from the surface to the deeper region at the level of the occipital gyri, reflecting only partial alignment with the manually marked electrode positions. The used terminology is adapted from Table 2, with the double-digit numbers being manually placed electrodes and the triple-digit numbers being electrodes placed via neuronavigation.

**Table 1 animals-14-01539-t001:** Definitions of the epileptogenic zone and the associated diagnostic techniques. Table adapted from [9,16,17].

Cortical Zone	Definition	Diagnostic Techniques
Epileptogenic zone	Region of cortex that can generate epileptic seizures and the removal or disconnection of which should lead to seizure freedom	Postoperative seizure outcome
Epileptogenic lesion	Distinct brain lesion, capable of generating and sustaining epileptic seizures	MRI
Symptomatogenic zone	Region of cortex that generates the initial seizure presentation (signs)	Seizure semiology (video; video-EEG)
Irritative zone	Region of cortex that generates inter-ictal epileptiform discharges on EEG	EEG; ECoG; MEG; EEG-triggered fMRI
Seizure-onset zone	Region where a clinical seizure originates	EEG; video-EEG; ECoG; (ictal SPECT; MEG)
Functional deficit zone	Region of cortex that, in the inter-ictal period, is clinically and/or electrophysiologically abnormal	Neurological exams; functional imaging (ictal SPECT; inter-ictal PET; functional MRIs)

Abbreviations: ECoG, electrocorticography; EEG, electroencephalography; fMRI, functional MRI; MEG, magnetoencephalography; PET, positron emission tomography; and SPECT, single-photon emission computed tomography.

**Table 2 animals-14-01539-t002:** Electrode terminology for recording canine electroencephalograms. Adaption of Table 1 from Pellegrino and Sica [22].

Fp1 and Fp2	Left and right frontopolar electrodes
F3 and F4	Left and right frontal electrodes
P3 and P4	Left and right parietal electrodes
O1 and O2	Left and right occipital electrodes
T3 and T4	Left and right temporal electrodes
Cz	Central vertex electrode
Oz	Middle occipital electrode

**Table 3 animals-14-01539-t003:** Anatomical sites suggested for the positioning of electroencephalographic recording electrodes in canines (electrodes must be located at the crossing of the transverse and sagittal or median planes described below, except for the temporalis electrodes). Adaption of Table 2 from Pellegrino and Sica [22].

Electrode	Mesocephalic and Dolichocephalic Head Type	Brachycephalic Head Type	Brain-Projecting Area
F	Transverse plane: draw an imaginary line about 0.5–1 cm cranial to the meeting of the temporal lines at midline	Transverse plane: draw an imaginary line that crosses the caudal margin of the zygomatic process of the frontal bone	Frontal lobe: precruciate gyrus
	Sagittal plane: quarter of the distance between the midline and the zygomatic arch	Sagittal plane: a quarter of the distance between the midline and the zygomatic arch	
Fp	Transverse plane: draw an imaginary line through the lateral canthus of the eye	Transverse plane: draw an imaginary line starting at half the distance between the lateral and medial canthus of the eye	Frontal lobe: prorean gyrus
	Sagittal plane: in line with the frontal electrodes	Sagittal plane: in line with the frontal electrodes	
O	Transverse plane: draw an imaginary line at the level of the mastoid process, at the base of the auricular part of the temporal bone	Transverse plane: draw an imaginary line at the level of the caudal margin of the ear base	Occipital lobe: marginal/occipital gyrus
	Sagittal plane: in line with the frontal electrodes	Sagittal plane: in line with the frontal electrodes	
P	Transverse plane: at half the distance between the frontal and the occipital electrodes	Transverse plane: at half the distance between the frontal and the occipital electrodes	Parietal lobe: rostral part of the ectomarginal gyrus
	Sagittal plane: in line with the frontal and occipital electrodes	Sagittal plane: in line with the frontal and occipital electrodes	
T	Having the reference of the dorsal edge of the caudal aspect of the zygomatic arch, just at the beginning of the temporal crest	Having the reference of the dorsal edge of the caudal aspect of the zygomatic arch, just at the beginning of the temporal crest	Temporal lobe: pseudosylvian fissure
Cz	At the midline, at the level of the parietal electrodes	At the midline, at the level of the parietal electrodes	Parietal area: longitudinal fissure
Oz	At the midline, at the level of the occipital electrodes	At the midline, at the level of the occipital electrodes	Occipital area: longitudinal fissure

**Table 4 animals-14-01539-t004:** Percentage and number of electrodes (n = 264) that are aligned, partially in alignment, or not aligned with the desired BPA for each manual-placement localization.

Alignment with BPA	Fp1	Fp2	F3	F4	P3	P4	O1	O2	T3	T4	Cz	Oz	Total
yes	0% (n = 0/22)	0% (n = 0/22)	41% (n = 9/22)	41% (n = 9/22)	68% (n = 15/22)	68% (n = 15/22)	73% (n = 16/22)	73% (n = 16/22)	27% (n = 6/22)	41% (n = 9/22)	100% (n = 22/22)	100% (n = 22/22)	53% (n = 139/264)
partial	41% (n = 9/22)	41% (n = 9/22)	23% (n = 5/22)	23% (n = 5/22)	0% (n = 0/22)	0% (n = 0/22)	23% (n = 5/22)	23% (n = 5/22)	0% (n = 0/22)	0% (n = 0/22)	0% (n = 0/22)	0% (n = 0/22)	14% (n = 38/264)
no	59% (n = 13/22)	59% (n = 13/22)	36% (n = 8/22)	36% n = 8/22)	32% (n = 7/22)	32% (n = 7/22)	5% (n = 1/22)	5% (n = 1/22)	73% (n = 16/22)	59% (n = 13/22)	0% (n = 0/22)	0% (n = 0/22)	33% (n = 87/264)
Total of not aligned	100% (n = 22/22)	100% (n = 22/22)	59% (n = 13/22)	59% (n = 13/22)	32% (n = 7/22)	32% (n = 7/22)	27% (n = 6/22)	27% (n = 6/22)	73% (n = 16/22)	59% (n = 13/22)	0% (n = 0/22)	0% (n = 0/22)	47% (n = 125/264)

Abbreviations: BPA, brain-projecting area; Fp1 and Fp2, left and right frontopolar electrodes; F3 and F4, left and right frontal electrodes; P3 and P4, left and right parietal electrodes; O1 and O2, left and right occipital electrodes; T3 and T4, left and right temporal electrodes; Cz, central vertex electrode; and Oz, middle occipital electrode.

**Table 5 animals-14-01539-t005:** Direction of deviation from the BPA and percentage/number of deviated electrodes for each manual electrode placement localization that did not align or only partially aligned with the required BPA. In total, 125 were not aligned with the BPA.

Deviation from BPA	Fp1	Fp2	F3	F4	P3	P4	O1	O2	T3	T4	Total
lateral	45% (n = 10/22)	45% (n = 10/22)	31% (n = 4/13)	31% (n = 4/13)							22% (n = 28/125)
lateral, rostral	55% (n = 12/22)	55% (n = 12/22)	54% (n = 7/13)	54% (n = 7/13)	14% (n = 1/7)	14% (n = 1/7)					32% (n = 40/125)
lateral, caudal			8% (n = 1/13)	8% (n = 1/13)	86% (n = 6/7)	71% (n = 5/7)	33% (n = 2/6)	33% (n = 2/6)			14% (n = 17/125)
caudal			8% (n = 1/13)	8% (n = 1/13)		14% (n = 1/7)	50% (n = 3/6)	50% (n = 3/6)	44% (n = 7/16)	31% (n = 4/13)	16% (n = 20/125)
caudal, dorsal									50% (n = 8/16)	62% (n = 8/13)	13% (n = 16/125)
rostral							17% (n = 1/6)	17% (n = 1/6)			2% (n = 2/125)
dorsal									6% (n = 1/16)	8% (n = 1/13)	2% (n = 2/125)
Total	100% (n = 22/22)	100% (n = 22/22)	59% (n = 13/22)	59% (n = 13/22)	32% (n = 7/22)	32% (n = 7/22)	27% (n = 6/22)	27% (n = 6/22)	73% (n = 16/22)	59% (n = 13/22)	47% (n = 125/264)

Abbreviations: BPA, brain-projecting area; Fp1 and Fp2, left and right frontopolar electrodes; F3 and F4, left and right frontal electrodes; P3 and P4, left and right parietal electrodes; O1 and O2, left and right occipital electrodes; T3 and T4, left and right temporal electrodes; Cz, central vertex electrode; and Oz, middle occipital electrode.

**Table 6 animals-14-01539-t006:** Cortical structures that the deviated manually placed electrodes overlayed, showing the percentage and number of electrodes which overlayed each structure for partially aligned and nonaligned electrodes. Partial alignment or no alignment meant that the electrode deviated from the following, desired brain-projection area and cortical structure: Fp electrodes—frontal lobe, precruciate gyrus; F electrodes—frontal lobe, prorean gyrus; O electrodes—occipital lobe, marginal/occipital gyrus; P electrodes—parietal lobe, rostral part of the ectomarginal gyrus; and T electrodes—temporal lobe, pseudosylvian fissure.

Electrode	Partial Alignment	Amount	No Alignment	Amount
Fp	Alignment only in the transverse plane	32% (n = 14/44)	No brain structure underlying	59% (n = 26/44)
	Position over precruciate/prorean gyrus not distinguishable	9% (n = 4/44)		
F	Alignment only in the transverse plane	23% (n = 6/26)	Postcruciate gyrus	15% (n = 4/26)
	Alignment only in the sagittal plane	8% (n = 2/26)	Prorean gyrus	38% (n = 10/26)
	Position over precruciate/prorean gyrus not distinguishable	8% (n = 2/26)	No brain structure underlying	8% (n = 2/26)
P			Medial part of the ectomarginal gyrus	71% (n = 10/14)
			Caudal part of the ectomarginal gyrus	14% (n = 2/14)
			Frontal lobe *: precruciate gyrus	14% (n = 2/14)
O	Alignment only in the sagittal plane	83% (n = 10/12)	Parietal lobe *: medial part of the ectomarginal gyrus	17% (n = 2/12)
T			Occipital lobe *: occipital gyrus	14% (n = 4/29)
			Ectosylvian gyrus	52% (n = 15/29)
			Composite gyrus	3% (n = 1/29)
			Ectomarginal gyrus	17% (n = 5/29)
			Sylvian gyrus	14% (n = 4/29)

* = A diversion not only to a different cortical structure but also to a different cortical lobe. Abbreviations: F, frontal electrodes; Fp, frontopolar electrodes; O, occipital electrodes; P, parietal electrodes; and T, temporal electrodes.

## Data Availability

The raw data supporting the conclusions of this article will be made available by the corresponding author upon request.

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
