# Peer review of "Canine Electroencephalography Electrode Positioning Using a Neuronavigation System"

_animals, 2024, doi:10.3390/ani14111539_

Round 1

Reviewer 1 Report

Comments and Suggestions for Authors

Dear authors,

The introduction sets the context well. Line 67, you could have added that there si (as far as I know) no literature on the correspondance between electrode positions and cerebral functional areas, as the aim is to explore the functioning of the cortex using EEG.

line 150 : Could you replace the word "seven" by "7" ?

Maybe could you say "with 11 (n=11/22) male and 11 female (n=11/12) female dogs "?

line 151 : Could you please replace the words "one", ",three", "seven" and "eight" by numbers ?

Paragraph "Alignment" : is the term marked electrode position the same as  "manually marked electrode position" or MPE ? If yes, could you use please MPE ?

The term variance ligne 196, 197 and 198 should perhaps be replaced to avoid confusion with statistical variance ?

Thank you for your discussion, which explains the dilemna between a EEG clinical pratice easy to set up and a EEG practice for experimental and surgical purposes that requires accurate electrode placement and a more complex set-up.

Author Response

Dear Reviewer,

We greatly appreciate your work with our manuscript. We will try to incorporate all your remarks into our revised manuscript, where possible.

Thank you very much and we hope you are pleased by the changes we’ve made.

Kind regards,

Casey Rogers, Nina Meyerhoff, Sebastian Meller, Holger Volk

Dear authors,

The introduction sets the context well. Line 67, you could have added that there si (as far as I know) no literature on the correspondance between electrode positions and cerebral functional areas, as the aim is to explore the functioning of the cortex using EEG.

Thank you for this comment. This is indeed an interesting observation. However, we felt that incorporating this may exceed the scope of this study, as one may discuss that (in humans at least) certain Brodmann areas may be assessed via EEG. As we did not further assess these correspondences we feel, it may cause confusion when adding this remark to the introduction and consequently the discussion. However of course we may not fully understand your query and are happy to discuss this further, should our response dissatisfy you.

line 150 : Could you replace the word "seven" by "7" ?

Thank you for this remark, due to Animals writing guidelines numbers 1-9 must be spelled out as words, so we unfortunately cannot change this.

Maybe could you say "with 11 (n=11/22) male and 11 female (n=11/12) female dogs "?

Thank you, we will change this sentence accordingly to make it clearer.

line 151 : Could you please replace the words "one", ",three", "seven" and "eight" by numbers ?

As mentioned above unfortunately this cannot be changed due to the guidelines.

Paragraph "Alignment"  is the term marked electrode position the same as  "manually marked electrode position" or MPE ? If yes, could you use please MPE ?

Yes, it is, thank you for noticing. We will adjust this accordingly.

The term variance ligne 196, 197 and 198 should perhaps be replaced to avoid confusion with statistical variance ?

Thank you for this important remark, we will replace the word variance with variability, to hopefully avoid confusion.

Thank you for your discussion, which explains the dilemna between a EEG clinical pratice easy to set up and a EEG practice for experimental and surgical purposes that requires accurate electrode placement and a more complex set-up.

Thank you very much for this kind comment.

Reviewer 2 Report

Comments and Suggestions for Authors

Overall, this is a relevant, novel, and well-written manuscript. The findings are interesting and prompt further research in this field.

Please find my comments below:

The introduction includes relevant citations, however most terms in table 1 are not used throughout the manuscript and therefore may be superfluous.

The material and methods are generally appropriate, although some clarification of a few details should be provided. The study aims to investigate the variability of EEG electrode positions to their underlying cortical region using a neuronavigation system and in the discussion, it is mentioned that the study examined whether a positioning of electrodes using the neuronavigation system improves accuracy. I suspect this means improving accuracy of electrodes placed using neuronavigation compared to the manually placed electrodes which in my understanding has not been assessed? Although it states under the head ‘evaluation’ that the alignment of the MPE has been evaluated with the desired BPA, this should also be made clear in the results, including the tables to avoid confusion. Perhaps a figure (similar to fig 4) may be helpful in the M&M to visualise how the variability has been measured. In addition, could you specify how complete/partial/non-alignment has been determined?

A suggestion, have you looked whether there is a difference between doliocephalic and brachycephalic dogs regarding electrode placement alignment? It may be interesting to see whether there are differences (even just descriptive) at, for example, the level of the frontal lobe.  

The results are otherwise clear, and the tables are very useful. The results contribute significantly to the research literature in this area of investigation.

The conclusion is clear, although accuracy has not been clearly defined in the materials and methods. I would also remove the last sentence of your conclusion as this is just a hypothesis, has not been evaluated by your study, and weakens the clinical significance of your research.

The reference list is relevant and complete. I could only not find reference 25. 

Comments on the Quality of English Language

The English language is overall well-employed with only minor changes necessary. Several areas are missing a comma. The dogs head should be changed to the dog's head or canine head. Be mindful of single and plural (e.g. singular gyri change to tentative gyrus). 'Fixated' should be 'fixed'. Page 4 of 19, line 242-244, the grammar in this sentence is incorrect. 

Author Response

Dear Reviewer,

We greatly appreciate your work with our manuscript. We will try to incorporate all your remarks into our revised manuscript.

Thank you very much for your kind remarks and we hope you are pleased by the changes we’ve made.

Kind regards,

Casey Rogers, Nina Meyerhoff, Sebastian Meller, Holger Volk

Overall, this is a relevant, novel, and well-written manuscript. The findings are interesting and prompt further research in this field. 

Please find my comments below: 

The introduction includes relevant citations, however most terms in table 1 are not used throughout the manuscript and therefore may be superfluous.

We understand that due to the lack of use of the terms, this table seems superfluous. With the changes made in this review we hope to have incorporated a few more of the terms, making table one a useful overview on important definitions. We found that a table more clearly summarizes the diagnostic techniques for said definitions and how EEGs play a role, than if we were to explain the used definitions in the main text.

The material and methods are generally appropriate, although some clarification of a few details should be provided. The study aims to investigate the variability of EEG electrode positions to their underlying cortical region using a neuronavigation system and in the discussion, it is mentioned that the study examined whether a positioning of electrodes using the neuronavigation system improves accuracy. I suspect this means improving accuracy of electrodes placed using neuronavigation compared to the manually placed electrodes which in my understanding has not been assessed?

We apologize for this inconvenience. Thank you for this important remark. We have adapted the results section to make this clearer. As mentioned in the discussion and briefly stated in figure 3, we did compare the two approaches. Hopefully our adjustment of the results section alleviates any future confusions. We also adapted the descriptions of the tables to make it clear, that those results refer to the manual placement.

Although it states under the head ‘evaluation’ that the alignment of the MPE has been evaluated with the desired BPA, this should also be made clear in the results, including the tables to avoid confusion. Perhaps a figure (similar to fig 4) may be helpful in the M&M to visualise how the variability has been measured.

Thank you for this recommendation, we will incorporate more details on how the variability is measured. We’ll also add that the MPE has been evaluated with the desired BPA in the results and tables to make it clearer and hopefully alleviate any confusions. We hope our changes address your concerns.

 In addition, could you specify how complete/partial/non-alignment has been determined?

Yes of course, we will specify this in the material and methods section. Thank you for this remark.

A suggestion, have you looked whether there is a difference between doliocephalic and brachycephalic dogs regarding electrode placement alignment? It may be interesting to see whether there are differences (even just descriptive) at, for example, the level of the frontal lobe.  

Thank you for this great input. We had discussed this for T electrodes (lines 326 ff). However, since this is a very interesting remark, we added additional information regarding the Fp electrodes. Additionally further details could also be found in Table 2 of the supplementary Material, where each individual dog is listed.

The results are otherwise clear, and the tables are very useful. The results contribute significantly to the research literature in this area of investigation.

Thank you very much for this friendly remark.

The conclusion is clear, although accuracy has not been clearly defined in the materials and methods. I would also remove the last sentence of your conclusion as this is just a hypothesis, has not been evaluated by your study, and weakens the clinical significance of your research.

Thank you for this recommendation, we will remove this part of the conclusion.

The reference list is relevant and complete. I could only not find reference 25. 

We apologize for this inconvenience. The reference was a brief presentation of Roberto Pomas (then) current studies. Unfortunately, since his work on the neuronavigation never got published we were only able to use this as a reference. The presentations held at the “Brain Camp” aren’t open access. Nevertheless, we felt the need to appropriately cite Poma, as his unpublished work was a key aspect of the foundation for the idea behind this manuscript. If this leaves any questions please do not hesitate to contact us.

Comments on the Quality of English Language

The English language is overall well-employed with only minor changes necessary. Several areas are missing a comma. The dogs head should be changed to the dog's head or canine head. Be mindful of single and plural (e.g. singular gyri change to tentative gyrus). 'Fixated' should be 'fixed'. Page 4 of 19, line 242-244, the grammar in this sentence is incorrect. 

Thank you for this remark and your corrections. We will adjust the spelling and grammar accordingly.